# Effects of social determinants of health on obesity among urban women of reproductive age

**Dickson A. Amugsi** [1] *, **Zacharie T. Dimbuene** [2,3]

**1** Nutrition and Food Systems Unit, African Population and Health Research Center, Nairobi, Kenya,
**2** Department of Population Sciences and Development, University of Kinshasa, Kinshasa, The Democratic Republic of the Congo, **3** Microdata Access Division, Statistics Canada, Ottawa, Canada

* damugsi2002@yahoo.com

**Data Availability Statement:** The datasets generated and/or analysed during the current study are available in the DHS program repository, https://dhsprogram.com/data/available-datasets. cfm. Data are accessible free of charge upon a

## Abstract

Obesity is a major global public health problem. It is spreading very fast in low- and middle-income countries and has reached world record levels in some of them. In Ghana, it has increased by over 65% among urban women in the past three decades. This study investigated the effects of social determinants of health on obesity among women in urban Ghana. The study analyzed the Ghana demographic and health survey data. These are nationally representative data collective every five years across low- and middle-income countries. A total of 1,204 urban women were included in the analysis. Body mass index was the outcome variable of interest. We used logistic regression to model the effects of the various social determinants of health on obesity. The results showed that 40% (95% confidence interval (CI) = 25.4, 57.0) and 36.7% (95% CI = 25.6, 49.3) of women who had higher education and those whose partners had higher education suffered from obesity, respectively. Women living in rich households had a five times higher prevalence of obesity than those in poor households (28.8% vs 5.7%). Further, 33.4% (95% CI = 18.5, 19.3) of women who occupied managerial positions were obese. The results from the multivariable logistic regression analysis suggested that compared to women in poor households, those in rich households were 3.4 times (95% CI = 1.31, 8.97) more likely to suffer from obesity. Women whose main occupation was agriculture were 81% (aOR = 0.19; 95% CI = 0.034, 0.99) less likely to suffer from obesity compared to those with no occupation. The results suggest that the various social determinants of health (SDH) included in the analysis significantly influence obesity outcomes. Women and partner education levels, occupying a managerial position, and living in rich households increase the risk of obesity. Interventions to address the rising obesity in urban Ghana should have specific packages targeted at these sub-groups.

## Introduction

Obesity (body mass index greater than or equal to 30kg/m2), an indicator of excessive fat accumulation in the body tissues, is a growing global problem of public health concern [1, 2]. Obesity is an illness that requires deliberate policies and interventions to address it to prevent

registration with the Demographic and Health Survey program (The DHS Program). The registration is done on the DHS website indicated above.

**Funding:** The authors received no specific funding for this work.

**Competing interests:** The authors have declared that no competing interests exist.

premature deaths among the world's population [2]. The available data suggest that since 1975, the global burden of obesity has more than tripled [3]. The data also suggest that no country has experienced a decline in obesity prevalence over this period. Therefore, it is unsurprising that in 2000, the World Health Organisation (WHO) declared obesity a pandemic, which informed the development of a global action plan to combat it [4]. Further evidence suggests that it is more severe among women of reproductive age than men and in urban settings than rural [3].

Obesity is spreading very fast in developing countries and has reached world record levels in some of them [5]. In Africa, the obesity prevalence among women of reproductive age ranged from 6.5 to 50.7% [6]. The problem is most severe among urban women relative to rural women [7]. Similarly, a recent analysis [2] using urban data from 24 African countries, spanning almost three decades, showed that all the countries included in the study experienced a substantial increase in obesity among women of reproductive age. Indeed, a country such as Ghana recorded 65% increase in obesity during the study period [2]. Other studies also observed alarming trends in obesity among women of reproductive age in the African region [8, 9]. There is a need for deliberate policies by governments to address the problem. However, it appears there is lack of policy prioritisation regarding obesity in Africa. This may be the case because many policymakers are yet to appreciate the seriousness of the problem. Therefore, it is critical to provide policymakers and other stakeholders with evidence of the seriousness of the problem and its potential drivers.

The consequences of obesity on population health are enormous. It is associated with an increased risk of cardiovascular diseases (mainly heart disease and stroke), diabetes, musculoskeletal disorders, and some cancers, including but not limited to endometrial, breast, ovarian, prostate, liver, gallbladder, kidney, and colon [10]. An analysis of recent data suggests that obesity is linked to 13 different cancers [11]. Also, the effect of obesity on the incidence of gestational diabetes, pre-eclampsia, high risk of miscarriage/stillbirth, and congenital anomalies is well documented [12–14]. Pregnant women who are obese have higher chances of heart disease, hypertension, and type 2 diabetes [15]. The link between women obesity and future obesity in their children has also been observed [16].

Similarly, the extant literature suggests a strong link between obesity and high mortality rate. An analysis of global data shows that 2.8 million people die annually due to obesity [17], while 35.8 million disability-adjusted life years (DALYs) are attributed to elevated body mass index (BMI) [15]. The adverse effects of obesity on population health, as outlined above, calls for a more detailed analysis to understand the determinants of the problem among women of reproductive age.

Previous research suggests that social determinants of health are key drivers of the rising obesity in urban Africa [18, 19]. According to the WHO commission, "*Social determinants of health (SDH) are the conditions in which people are born, grow, live, work and age, and the wider set of forces and systems shaping the conditions of daily life" [3].* The SDH includes socioeconomic status, education, literacy, neighborhood and physical environment, employment, food consumption, place of residence and social support networks, and access to health care. The literature on the effects of socioeconomic status, education, and employment status, among others, abound [20]. However, the impact varies between low- and high-income countries. While obesity is a significant issue among the rich in low-income countries, it is high among the poor in high-income countries [21].

Furthermore, obesity was more prevalent among highly educated women than those without formal education in low-income countries [22]. Women who are formally employed have increased odds of obesity compared to those who are not employed, which vary by educational attainment [23]. This could be attributed to increased income and behavioral changes related

to the time allocated to physical activity. In addition, the existing literature suggests a link between urbanization, nutritional transition, and the high prevalence of obesity among women [24]. There have been increasing changes in food consumption, whereby urban dwellers now consume more processed and fast foods and sugary beverages, with consequential adverse effects on obesity outcomes [24, 25].

Although our previous analysis suggests that obesity among women of reproductive age has been rising rapidly in urban Ghana in the past decades [2], there is limited knowledge on the key SDH driving the surge. The present study intends to fill this gap by undertaking a comprehensive analysis of obesity prevalence by various SDH, and the effects of the SDH on the rising obesity in urban Ghana, drawing on existing data sources. This type of analysis is needed to provide detailed and nuanced findings for better decision-making to inform policy and program interventions to address the obesity challenges in urban Ghana. The present study examined the effects of SDH on obesity among women of reproductive age in urban Ghana using existing data.

## Materials and methods

### Data sources and study design

The study is a secondary analysis of the 2014 Ghana Demographic and Health Surveys (DHS) data [26]. These are nationally representative data collected every five years in Ghana and other low- and middle-income countries (LMICs) [26]. The analysis in the current study is restricted to the urban sub-sample. We downloaded the data from the DHS program website and assessed their completeness regarding the women's anthropometric data. We excluded cases that were missing anthropometric data.

The DHS utilizes a complex sampling design, whereby the sample selection involves multiple stages. In the first stage, clusters are selected from a master sampling frame constructed from the Ghana National Population and Housing Census data. The clusters are selected using systematic sampling with probability proportional to size. After selecting the clusters, a household listing operation is undertaken in the selected clusters to get a sampling frame to select households to take part in the study. A systematic sampling of households is then undertaken. This stage of selection is aimed to ensure an adequate sample size to estimate the indicators of interest with acceptable precision. Our analyses are restricted to adult non-pregnant women aged 15–49 years. We focus on non-pregnant women because pregnancy is usually associated with weight gain. Therefore, including them in the analysis may present a misleading picture of the obesity situation. The eligible sample size used in the present analysis was 1,204 urban women of reproductive age.

### Ethics statement

The DHS study was undertaken based on high ethical standards. The data collectors were trained to respect the rights of study participants. The participants were made to understand that they have the right to decide whether they wanted to participate in the study or not, as well as to abstain or withdraw their participation at any time without reprisal. The potential risks and benefits associated with the study and steps taken to mitigate potential risks were adequately explained to study participants. A written informed consent was obtained from the parent/guardian of each participant under 18 years of age. A government-recognized institutional review committee granted ethical approval for the conduct of the study. Further ethical clearance was granted by the Institutional Review Board of ICF International, USA before the survey was conducted. The authors obtained permission from DHS Program for the use of the data. The authors did not seek further ethical clearance as the DHS data are highly anonymised.

## Measures

**Outcome variable.** The women's body mass index (BMI) was the outcome variable of interest in this analysis. Well-trained field technicians collected participants anthropometric data (Height and weight) using recommended techniques [27, 28]. The weight of study participants was measured using electronic Seca scales, while their height was measured using boards produced by Shorr Productions. These anthropometric data were used to estimate the BMI of the study participants. The BMI was calculated by dividing weight in kilograms by the square of height in meters. Based on the WHO guidelines [27], obesity was classified as BMI≥30.0 kg/m2. The prevalence of obesity and its associated factors were estimated. In the DHS dataset, place of residence is classified into rural and urban. However, this study focuses only on urban settings. Our previous analysis [2] informed the choice of the study setting. Also, a place of resident is a critical SDH, therefore, restricting the analysis to the urban settings will help to contextualize the study findings.

**Explanatory variables.** The SDH factors used in this analysis included women's education level, employment status, fruit and vegetable consumption, partner occupation, partner education level, household wealth index, number of trips in the last 12 months, frequency of listening to radio, and sex of household head. Potential control variables included women's age, height, and breastfeeding status. The household wealth index, a composite measure of the household cumulative living standard, was constructed using principal component analysis (PCA) [29]. The variables used in the PCA included but not limited to household ownership of televisions and bicycles, materials used for housing construction, and types of water access and sanitation facilities. The constructed wealth index is then divided into quintiles (poorest, poor, middle, rich & richest). To preserve the sample size for the analysis, we recoded the wealth index into poor (poorest+poor), middle, and rich (rich+richest). The potential SDH outlined above were identified based on the SDH literature. These variables were subjected to bivariate analysis to establish those associated significantly with women's obesity. Statistical significant variables were included in the multivariable analysis. In addition, we included variables that were not statistically significant but considered critical as far as obesity was concerned.

## Analytical strategy

The data analysis was performed using STATA V.14. The analyses involved three stages. In the first analysis, we assessed the characteristics of the sample using frequencies and means. In the second stage, we estimated the prevalence of obesity and their associated 95% confidence intervals (CIs) by various SDH. In the third analysis, we examined the effects of SDH on maternal obesity using multivariable logistic regression. We entered SDH variables in the first model (model 1), and adjusted for women's age, height, and breastfeeding status in the second model (model 2). Adjusted odds ratio (aOR) and CIs are reported for this analysis. We accounted for the complex survey design (CSD) effect in all analyses using the *svyset* and *svy* procedures in STATA.

## Results

### Background characteristics of the study sample

Table 1 presents the characteristics of the study sample. The average age of study participants was 31 years (30.85±6.29), while the average height was 1.59m (1.59±0.06). Majority (59%) of the women had secondary school education, while only 6% had tertiary/higher education. This was similar to the partner education level where 59% had secondary education. Similarly, 52%

**Table 1. Characteristics of the sample (n = 1204).**

| Variables | Mean±SD or % |
| --- | --- |
| **Maternal education** | |
| No education | 16.78 |
| Primary | 17.43 |
| Secondary | 59.48 |
| Higher | 6.31 |
| **Maternal occupation** | |
| No occupation | 21.63 |
| Professional/tech/managerial | 5.99 |
| Clerical/sales/services | 51.85 |
| Agricultural worker | 6.2 |
| Manual worker | 14.32 |
| **Days ate fruits in the last 7 days** | |
| None | 18.25 |
| 1–3 days | 39.8 |
| 4+ days | 41.95 |
| **Days ate vegetables in the last 7 days** | |
| None | 16.76 |
| 1–3 days | 38.58 |
| 4+ days | 44.67 |
| **Partner occupation** | |
| Manual worker | 45.41 |
| Professional/tech/managerial | 17.25 |
| Clerical/sales/services | 17.14 |
| Agricultural worker | 11.33 |
| **Sex of household head** | |
| Male | 68.23 |
| Female | 31.77 |
| **Partner education level** | |
| No education | 12 |
| Primary | 6.12 |
| Secondary | 58.92 |
| Higher | 14.65 |
| Maternal mean age | 30.85±6.29 |
| Maternal mean height | 1.59±0.06 |

of the women had clerical/sales/services as their main occupation. However, the dominant occupation of their partners was manual work (45%). The results also suggested that over 40% indicated they consumed fruits (42%) and vegetables (45%) for 4 or more days in the past 7 days.

Table 2 shows the prevalence of obesity by various SDH. The results showed that 40% (95% confidence interval (CI) = 25.4, 57.0) of women who had higher education were obese, compared to only 14.4% (95% CI = 8.6, 23.0) of those with no formal education. Also, 36.7% (95% CI = 25.6, 49.3) of women whose partners had higher education were obese. The prevalence of obesity was 28.8% (95% CI = 24.1, 34.0) among women living in rich households compared to 5.7% (95% CI = 2.4, 13.0) of those in poor households. Further, 33.4% (95% CI = 18.5, 19.3) of women who occupied managerial positions were obese. Conversely, only 1.8% (95% CI = 0.4, 7.8) of those whose occupation was agriculture suffered from obesity. Women whose partners

**Table 2. Prevalence of obesity by social determinants of health (n = 1204).**

| | Obese | | |
|---|---|---|---|
| *Variables* | *%* | *95% CI* | *P-value* |
| **Women's education** | | | |
| No education | 14.4 | 8.6, 23.0 | 0.026 |
| Primary | 25.4 | 17.2,35.7 | |
| Secondary | 23.5 | 18.6, 29.3 | |
| Higher | 40.2 | 25.4, 57.0 | |
| **Household wealth index (HWI)** | | | |
| Poor | 5.7 | 2.4, 13.0 | 0.001 |
| Middle | 13.0 | 8.5, 19.3 | |
| Rich | 28.8 | 24.1, 34.0 | |
| **Women's occupation** | | | |
| No occupation | 14.5 | 9.1, 22.2 | 0.001 |
| Managerial/technical | 33.4 | 18.5, 52.5 | |
| Clerical/sales/services | 27.9 | 23.2, 33.3 | |
| Agricultural worker | 1.8 | 0.4, 7.8 | |
| Manual worker | 25.2 | 16.2, 37.1 | |
| **Number of days ate fruits** | | | |
| None (zero days) | 13.6 | 7.7, 22.7 | 0.028 |
| ate fruits 1–3 days | 23.2 | 17.9, 29.6 | |
| Ate fruits 4+ days | 27.7 | 21.9, 34.4 | |
| **Number of days ate vegetables** | | | |
| None (zero days) | 21.2 | 13.2, 32.3 | 0.76 |
| Ate vegetables 1–3 days | 22.4 | 16.0, 30.4 | |
| Ate vegetables 4+ days | 25.0 | 19.7, 31.1 | |
| **Partner's occupation** | | | |
| Manual worker | 25.6 | 20.4, 31.6 | 0.001 |
| Managerial/technical | 33.6 | 24.9, 43.6 | |
| Clerical/sales/services | 24.7 | 16.6, 34.9 | |
| Agricultural worker | 7.2 | 3.3, 15.0 | |
| **Sex of household head** | | | |
| Male | 24.2 | 19.7, 29.4 | 0.57 |
| Female | 21.4 | 14.6, 30.3 | |
| **Partner education level** | | | |
| No education | 16.7 | 10.3, 25.8 | 0.001 |
| Primary | 8.2 | 3.6, 17.6 | |
| Secondary | 24.9 | 20.3, 30.1 | |
| Higher | 36.7 | 25.6, 49.3 | |
| **Women's literacy** | | | |
| Cannot read at all/no information/virtually impaired | 18.6 | 14.3, 24.0 | 0.072 |
| Able to read only part of the sentence | 32.0 | 17.6, 50.9 | |
| Able to write a whole sentence | 27.1 | 21.3, 33.9 | |
| **Frequency listen radio** | | | |
| Not at all | 11.6 | 6.5, 19.9 | 0.027 |
| Less than once a week | 26.1 | 19.1, 34.5 | |
| At least once a week | 25.5 | 20.2, 31.7 | |
| **Is breastfeeding** | | | |
| No | 27.6 | 22.3, 33.6 | 0.031 |
| Yes | 19.7 | 15.2, 25.1 | |

held managerial positions suffered more from obesity than those whose partners were agricultural workers (33.6 vs. 7.2%). All the above results were statistically significant.

Table 3 presents the effects of SDH on obesity among women. The results showed that compared to women in the poor households, those in the rich households were 3.4 times (95% CI = 1.31, 8.97) more likely to suffer from obesity. Women whose main occupation was agriculture were 81% (aOR = 0.19; 95% CI = 0.034, 0.99) less likely to suffer from obesity than those with no occupation. Similarly, women whose partners' occupation was agriculture worker had 71% (aOR = 0.39; 95% CI = 0.15, 0.99) reduced odds of becoming obese compared to those whose partners were manual workers. Partners attaining primary education had a moderate protective effect on obesity among women (aOR = 0.34; 95%CI = 0.11, 1.08). Compared to women who did not listen to radio at all, those who listened to radio less than once a week or at least once a week had increased odds of obesity (aOR = 2.64, 95% CI = 1.26, 5.54). A biological variable such as women's age was associated with increased odds of obesity (aOR = 1.09; 95% CI = 1.06, 1.13).

## Discussion

The extant literature shows that social determinants of health (SDH), as captured in the WHO commission on SDH framework, play a critical role in determining obesity and health outcomes [3, 20–23]. In this study, we presented a comprehensive analysis of obesity prevalence by various SDH and their effects on the rising obesity in urban Ghana, using existing data. The results suggest that obesity was three times higher among women with high education relative to those without education (40% vs 14.4%). This is also the case with partner education, where women whose partners were highly educated suffer more from obesity. The findings reflect the results of previous research. Several studies have documented the relationship between the level of education and the high prevalence of obesity [22, 23, 30, 31]. High education appears to place women at an elevated risk of obesity. The negative effect of education on obesity may be because this subgroup is likely to have a higher income, with a consequential impact on lifestyle changes. For example, the extant literature reveals that high-income earners in LMICs practice sedentary lifestyles and consume unhealthy diets [24, 30]. It is essential to point out that while education is a risk factor for obesity in LMICs, it is indeed a protective factor in high-come settings [21]. Thus, the highly educated in high-income countries are less prone to obesity than the less educated.

The analysis of the women's occupation data reveals that women holding managerial positions had a substantially higher prevalence of obesity than other occupations. The high prevalence may be because of prolonged sitting hours, reduced walkability, increased intake of convenient, highly processed foods, and greater reliance on cars as the primary means of transport [30, 32]. For example, long hours at the office can make it challenging to squeeze in time for exercise [33]. The finding in the present study is consistent with the extant literature. A study undertaken among Korean women showed that obesity prevalence was significantly higher among managers than non-managers [32].

Further, our results suggest that women who engage in agriculture suffer less from obesity compared to the other occupations. This could be attributed to the high physical activity associated with this occupation. This finding is consistent with the multivariable logistic regression results, which revealed that women who engaged in agriculture had an 81% reduced odds of becoming obese than those in other occupations. The low risk associated with agriculture may be linked to agricultural workers consuming their produce rather than relying on high-energy food such as fast food and sugar-sweetened beverages [32].

**Table 3. Multivariable logistic regression analysis of the association between social determinants of health and women obesity (n = 1204).**

| VARIABLES | Model 1 | Model 2 |
|---|---|---|
| **Women's education** | | |
| No Education (ref) | | |
| Primary | 1.624 | 1.776 |
| | (0.727–3.627) | (0.767–4.113) |
| Secondary | 0.937 | 0.914 |
| | (0.423–2.076) | (0.395–2.113) |
| Higher | 1.481 | 1.565 |
| | (0.408–5.382) | (0.418–5.861) |
| **Household wealth index** | | |
| Poor (ref) | | |
| Middle | 1.656 | 1.853 |
| | (0.637–4.306) | (0.698–4.920) |
| Rich | 3.262** | 3.421** |
| | (1.219–8.729) | (1.305–8.966) |
| **Women's occupation** | | |
| No occupation (ref) | | |
| Managerial/technical | 1.371 | 1.096 |
| | (0.429–4.380) | (0.334–3.601) |
| Clerical/sales/services | 2.088** | 1.626 |
| | (1.119–3.895) | (0.841–3.145) |
| Agricultural worker | 0.267 | 0.185** |
| | (0.055–1.298) | (0.034–0.989) |
| Manual worker | 1.870 | 1.354 |
| | (0.833–4.200) | (0.595–3.082) |
| **Partner's occupation** | | |
| Manual (ref) | | |
| Managerial/technical | 0.998 | 0.985 |
| | (0.474–2.102) | (0.449–2.160) |
| Clerical/sales/services | 0.821 | 0.772 |
| | (0.449–1.501) | (0.403–1.479) |
| Agricultural worker | 0.497 | 0.386** |
| | (0.197–1.250) | (0.150–0.992) |
| **Days ate fruits in the last 7 days** | | |
| None (ref) | | |
| 1–3 days | 1.585 | 1.667 |
| | (0.777–3.233) | (0.782–3.554) |
| 4+ days | 2.047** | 1.846* |
| | (1.097–3.820) | (0.935–3.647) |
| **Days ate vegetables in the last 7 days** | | |
| None (ref) | | |
| 1–3 days | 1.030 | 0.985 |
| | (0.489–2.168) | (0.447–2.170) |
| 4+ days | 1.015 | 0.958 |
| | (0.534–1.928) | (0.491–1.871) |
| **Partner's education level** | | |
| No education (ref) | | |

*(Continued)*

**Table 3.** (Continued)

| VARIABLES | Model 1 | Model 2 |
|---|---|---|
| Primary | 0.296** | 0.339* |
| | (0.093–0.941) | (0.107–1.075) |
| Secondary | 0.806 | 0.936 |
| | (0.378–1.716) | (0.425–2.065) |
| Higher | 1.019 | 1.234 |
| | (0.327–3.170) | (0.381–3.997) |
| **Number of trips in last 12 months (continuous)** | 1.015 | 1.015 |
| | (0.987–1.044) | (0.990–1.040) |
| **Frequency listening radio** | | |
| Not at all (ref) | | |
| Less than once a week | 2.747*** | 2.640** |
| | (1.345–5.609) | (1.258–5.540) |
| At least once a week | 2.268** | 2.295** |
| | (1.170–4.397) | (1.190–4.426) |
| **Women's age (in single years) (continuous)** | | 1.090*** |
| | | (1.055–1.126) |
| **Women's height (in meters)** | | 0.092 |
| | | (0.005–1.806) |
| **Sex of household head** | | |
| Male (ref) | | |
| Female | | 1.059 |
| | | (0.605–1.852) |
| **Breastfeeding status (Ref.: No)** | | |
| No (ref) | | |
| Yes | | 0.758 |
| | | (0.486–1.183) |

95% Confidence intervals in parentheses

*** $p < 0.01$

** $p < 0.05$

* $p < 0.1$

Similarly, the findings show that the prevalence of obesity among women living in rich households is over five times higher than those in poor households (28.8% vs 5.7%). It implies that wealth is a risk factor for women's obesity, while poverty is a protective factor. This finding has been corroborated by the multivariable analysis, where women in rich households were over three times more likely to suffer from obesity than those in poor households. The endemicity of obesity in rich households may be due to access to high-energy food and a lack of physical activity. This finding is consistent with the existing literature, whereby several studies have shown that women in rich households suffer more from obesity than those in poor households [34–37]. However, a previous comparative analysis showed some differences between low-and high incomes countries. Indeed, while obesity is a major public health issue among the rich in low-income countries, it is the case among the poor in high-income countries [21]. It may mean that the rich in the high-income countries are more concerned about their weight and therefore live healthy lifestyles.

### Strength and limitations of the study

The study used robust nationally representative data, making it possible to generalise the findings to all women of reproductive age living in urban settings in Ghana. The large sample size enhances the precision of the estimates and the robustness of the associations observed in the analysis. Further, the outcome variable was objectively measured, reducing the chances of misclassification. A noteworthy limitation is the cross-sectional nature of the data, which does not make it possible to establish any causal relationship between the various SDH and obesity. Therefore, the results are interpreted in the context of associations between the independent and dependent variables. The data did not allow the inclusion of all the broader social determinants of health as defined by the WHO commission on SDH framework. However, the SDH framework did not envision that a single study could address all the factors captured in the framework.

## Conclusions

The study examined the effects of the various social determinants of health on obesity among women of reproductive age in urban Ghana. The results suggest that the different SDH substantially influence women's obesity. Higher education increases the risk of obesity among women. Women occupying managerial positions are also likely to suffer from obesity. Similarly, living in rich households increases the risk of obesity among women. Interventions to address the rising obesity in urban Ghana should have specific packages targeted at these sub-groups. These may include promoting physical activity, and healthy eating behaviours, which the extant evidence suggests are hardly practised by these sub-groups.

## Acknowledgments

We wish to express our profound gratitude to The DHS Program, USA for providing us access to the data. We also wish to acknowledge institutions of respective countries that played critical roles in the data collection process.

## Author Contributions

**Conceptualization:** Dickson A. Amugsi.

**Data curation:** Dickson A. Amugsi, Zacharie T. Dimbuene.

**Formal analysis:** Dickson A. Amugsi, Zacharie T. Dimbuene.

**Methodology:** Dickson A. Amugsi.

**Writing – original draft:** Dickson A. Amugsi.

**Writing – review & editing:** Dickson A. Amugsi, Zacharie T. Dimbuene.

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
