## [Decision Letter · Decision Letter 0]

28 Nov 2022

PGPH-D-22-01537

Effects of social determinants of health on obesity among urban women of reproductive age

Dear Dr. Amugsi,

Thank you for submitting your manuscript to PLOS Global Public Health. After careful consideration, we feel that it has merit but does not fully meet PLOS Global Public Health’s publication criteria as it currently stands. Therefore, we invite you to submit a revised version of the manuscript that addresses the points raised during the review process.

We look forward to receiving your revised manuscript.

Kind regards,

Hasanain Faisal Ghazi, phd

Academic Editor

Journal Requirements:

1. In the online submission form, you indicated that your data will be submitted to a repository upon acceptance.  We strongly recommend all authors deposit their data before acceptance, as the process can be lengthy and hold up publication timelines. Please note that, though access restrictions are acceptable now, your entire data will need to be made freely accessible if your manuscript is accepted for publication. This policy applies to all data except where public deposition would breach compliance with the protocol approved by your research ethics board. If you are unable to adhere to our open data policy, please kindly revise your statement to explain your reasoning and we will seek the editor's input on an exemption. Please be assured that, once you have provided your new statement, the assessment of your exemption will not hold up the peer review process.

Additional Editor Comments (if provided):

respond to reviewers' comments

Reviewers' comments:

Reviewer's Responses to Questions

**Comments to the Author**

1. Does this manuscript meet PLOS Global Public Health’s publication criteria? Is the manuscript technically sound, and do the data support the conclusions? The manuscript must describe methodologically and ethically rigorous research with conclusions that are appropriately drawn based on the data presented.

Reviewer #1: Yes

Reviewer #2: Yes

2. Has the statistical analysis been performed appropriately and rigorously?

Reviewer #1: Yes

Reviewer #2: Yes

3. Have the authors made all data underlying the findings in their manuscript fully available (please refer to the Data Availability Statement at the start of the manuscript PDF file)?

Reviewer #1: Yes

Reviewer #2: Yes

4. Is the manuscript presented in an intelligible fashion and written in standard English?

Reviewer #1: Yes

Reviewer #2: Yes

5. Review Comments to the Author

Reviewer #1: Firstly in abstractno need to put BMI definition in abstract

In method: Why put low and middle income countries,you mean this is multi countries survey

SDH is what?

Introduction is too long , summerize

Study aim is not clear please clarify

Current analysis period (years)?

Only 6% agricultural so how come in conclusion work in agriculture is protective?

How wealth index categorized?

Table 3 R2 of the model?

Model 1 & Model 2 method of logestic regression?

Strength is 1.402 large enough?

Reviewer #2: The author raises important issues that need to be explored the evidence by research.

The evidence generates importance in providing an appropriate implementation in the reduction of obesity-related morbidity and mortality.

However, two amendments are needed to improve the quality of the research.

1. Try to analyses Minimum dietary diversity for women instead of considering only fruit and vegetable consumption.

2. Try to reconsider the relationship b/n higher education and obesity. Find deep review and justify how higher education is negative corelated obesity?

6. PLOS authors have the option to publish the peer review history of their article (what does this mean?). If published, this will include your full peer review and any attached files.

**Do you want your identity to be public for this peer review?** For information about this choice, including consent withdrawal, please see our Privacy Policy.

Reviewer #1: No

Reviewer #2: No

---

## [Editor Report · Decision Letter 1]

28 Dec 2022

Effects of social determinants of health on obesity among urban women of reproductive age

PGPH-D-22-01537R1

Dear Dr. Amugsi,

We are pleased to inform you that your manuscript 'Effects of social determinants of health on obesity among urban women of reproductive age' has been provisionally accepted for publication in PLOS Global Public Health.

Best regards,

Hasanain Faisal Ghazi, phd

Academic Editor